# Up-Regulation of PSMA Expression In Vitro as Potential Application in Prostate Cancer Therapy

**DOI:** 10.3390/ph16040538

**Published:** 2023-04-04

**Authors:** Roswitha Runge, Anne Naumann, Matthias Miederer, Joerg Kotzerke, Claudia Brogsitter

**Affiliations:** 1Department of Nuclear Medicine, University Hospital Carl Gustav Carus, Technical University Dresden, Fetscherstrasse 74, D-01307 Dresden, Germany; 2National Center for Tumor Diseases (NCT), D-01307 Dresden, Germany

**Keywords:** prostate cancer, PSMA, Lu-177–PSMA-617, PC3-PSMA, LNCaP, 5-aza-2′-deoxycitidine, valproic acid

## Abstract

Possibilities to improve the therapeutic efficacy of Lu-177–PSMA-617 radionuclide therapy by modulation of target expression are being investigated. Knowledge on regulatory factors that promote prostate cancer (PCa) progression may contribute to targeting prostate cancer more effectively. We aimed at the stimulation of PCa cell lines using the substances 5-aza-2′-deoxycitidine (5-aza-dC) and valproic acid (VPA) to achieve increased prostate-specific membrane antigen (PSMA) expression. PC3, PC3-PSMA, and LNCaP cells were incubated with varying concentrations of 5-aza-dC and VPA to investigate the cell-bound activity of Lu-177–PSMA-617. Stimulation effects on both the genetically modified cell line PC3-PSMA and the endogenously PSMA-expressing LNCaP cells were demonstrated by increased cellular uptake of the radioligand. For PC3-PSMA cells, the fraction of cell-bound radioactivity was enhanced by about 20-fold compared to that of the unstimulated cells. Our study reveals an increased radioligand uptake mediated by stimulation for both PC3-PSMA and LNCaP cell lines. In perspective of an enhanced PSMA expression, the present study might contribute to advanced radionuclide therapy approaches that improve the therapeutic efficacy, as well as combined treatment options.

## 1. Introduction

The prostate-specific membrane antigen (PSMA) is a membrane glycoprotein that presents an extracellular target, which is expressed on the surface of the prostate epithelial cells. As PSMA is highly expressed in prostate cancer (PCa) and also upregulated in metastatic, hormone-refractory carcinomas, it is an excellent target for imaging and therapy in nuclear medicine. PSMA-targeted ligands have been developed and are already established in diagnostics [1,2,3,4] and therapy of the metastatic castration-resistant prostate cancer (mCRPC). Therapeutic administration of the Lu-177-labeled PSMA-targeted ligands Lu-177-PSMA I&T [5] as well as Lu-177–PSMA-617 [6] was developed in the past. An international phase three trial conducted by Sartor et al. revealed that Lu-177–PSMA-617 prolonged progression-free overall survival in the case of combined treatment with the standard care of PCa [7].

PCa is a complex disease that is mediated by the accumulation of genetic and epigenetic changes, such as reorganization of the chromatin structure, differential expression of oncogenes and tumor-suppressor genes, or initiation of apoptosis [8]. Watt et al. developed a PSMA enhancer (PSME) that strongly activates the folate hydrolase-1 (FOLH1), the PSMA gene promoter region in prostate cell lines [9].

Epigenetic alterations, defined as heritable control in gene expressions in the absence of changes in DNA sequences, appear to contribute to the malignant transformation and progression of cancer [10]. Thus, a variety of research projects focus on the epigenetic fundamental background [11,12,13]. A better understanding of the epigenetic factors that promote prostate cancer progression may lead to the design of rational therapeutic strategies to target prostate cancer more effectively. The systematical review by Kumaraswamy et al. examines the role of epigenetic factors in prostate cancer [14]. A panel of genetic and epigenetic markers of prostate tumor cells resistance were identified by Kutilin et al. [15].

Investigations on the epigenetic stimulation of the SST2 receptor expression indicated increased cell-binding of the SST2 targeting radioligand Lu-177–DOTATATE in neuroendocrine tumor cells [16,17,18] as well as in a pre-clinical setting [19]. Although first clinical trials focusing on increasing the SST2 receptor expression exist, this issue needs further exploration [20,21].

In a recent study, we reported the successful enhancement of the SSTR2 expression in HEKsst_2_ cells by the epigenetic-like stimulation using the DNA methyltransferase inhibitor (DNMTi) 5-aza-2′-deoxycytidine (5-aza-dC) and the histone deacetylase inhibitor (HDACi) valproic acid (VPA). As a consequence of that and the deficiency of studies on the epigenetic PSMA stimulation, our motivation was to prove the concept of PSMA upregulation using cell lines that either express the PSMA ligand (PC3-PSMA, LNCaP) or show no PSMA expression (PC3). The aim of the study was to stimulate the PCa cell lines with the epigenetically efficient substances 5-aza-dC and VPA to prove the increased PSMA expression by increased cell-bound activity (uptake) of Lu-177–PSMA-617.

## 2. Results

### 2.1. Immunohistochemical Detection of PSMA Expression

PC3, PC3-PSMA, and LNCaP cells were stained with a fluorescence-labeled anti-human PSMA (FOLH1) antibody. The immunohistochemical staining demonstrates PSMA expressions in the cell membranes of PC3-PSMA and LNCaP, but not in PC3 cells. (Figure 1). Cell nuclei were counterstained with 4,6′-diamidino-2-phenylindole (DAPI). Hereby, the presence of the PSMA protein was confirmed on the cell membranes of PC3-PSMA and LNCaP cells, but not on PC3 cells.

### 2.2. Uptake of Lu-177–PSMA-617 in Dependency on 5-aza-dC and VPA Stimulation

Before uptake analysis, both cell lines were pre-treated with 5-aza-dC and VPA over a time period of 5 days as described in 4.3. Unstimulated controls were only incubated with Lu-177–PSMA-617.

Both 5-aza-dC and VPA reveal increased total cell-bound activity for all tested concentrations compared to the unstimulated PC3-PSMA and LNCaP cells. Non-stimulated LNCaP cells (control) show higher uptake values than non-stimulated PC3-PSMA cells. However, the combined pre-treatments with 1 µM 5-aza-dC/1 mM VPA or 5 µM 5-aza-dC/3 mM VPA result in the highest uptake levels, applicable for both PC3-PSMA and LNCaP cells. For example, in the case of the combined stimulation (5 µM 5-aza-dC/5 mM VPA), the Lu-177–PSMA-617 uptake after 1 h increased in PC3-PSMA by a factor of 21 (1.40 kBq/2.5 × 10^5^ cells ± 0.64 vs. 29.44 kBq/2.5 × 10^5^ cells ± 1.89) compared to non-stimulated control cells (*p* = 0.005); for LNCaP cells, the stimulation factor is 4.7 (4.79 kBq/2.5 × 10^5^ cells ± 0.47 vs. 22.52 kBq/2.5 × 10^5^ cells ± 6.45) *p* = 0.1113 (Figure 2b,c). Further, a statistically significant difference between the combined incubation with 1 µM 5-aza-dC/1 mM VPA and the unstimulated controls was found for both PC3-PSMA (1 h: *p* = 0.0229) and LNCaP (1 h: *p* = 0.0018).

Stimulated PC3 cells show no increase in the cell-bound activity compared to unstimulated cells after 1 h (Figure 2a). As PC3 cells served as negative control cells expressing only very small amounts of PSMA, the uptake was limited to the incubation time of 1 h.

### 2.3. Time Dependence of Lu-177–PSMA-617 Uptake in PC3, PC3-PSMA, and LNCaP Cells

Figure 3 shows a clear increase in the total cell-bound activity after 30 min for the cell lines PC3-PSMA and LNCaP. The LNCaP cells reach a maximal ligand uptake after 2 h (17.02 kBq ± 0.56) in contrast to PC3-PSMA cells achieving 10.33 kBq ± 0.31 ligand uptake after 2 h (*p* < 0.0001). The maximum uptake values are reached after 4 h incubation time for both LNCaP and PC3-PSMA cells, indicating a statistically significant difference (*p* = 0.0213). Intracellular accumulated Lu-177–PSMA-617 remains almost completely bound over 24 h. After 48 h, a slight decrease in the ligand uptake is visible, which is somewhat stronger pronounced in PC3-PSMA cells (*p* = 0.0008). In contrast, the low uptake values of PC3 at 4 h (0.1 kBq ± 0.01) cells remain unmodified over the time course of 48 h.

## 3. Discussion

In this short communication, we investigated the possible PSMA upregulation in prostate carcinoma cells using 5-aza-dC and VPA, already tested for their ability to stimulate the SST2 receptor expression [16]. The main issue is the target enlargement by enhancement of PSMA-ligand-binding sites on tumor cell surfaces. As a consequence, this approach can be an option to improve the therapeutic efficacy of Lu-177–PSMA-617 radionuclide therapy, especially for patients suffering from metastatic castration-resistant prostate cancer (mCRPC) that expresses low levels of PSMA.

Other studies also focusing on epigenetic-based concepts have been reported for SSTR2 in vitro [17,18] and in a pre-clinical setting [19].

As, up to now, very few studies have been performed on epigenetic-mediated PSMA expression, the interpretation of our results from the perspective of previous studies is partly restricted to results found for the unstimulated cell lines (without 5-aza-dC, VPA).

Our results show that the PSMA expression of the genetically modified cell line PC3-PSMA can be stimulated with the same drugs as the LNCaP cells, which are a typical PSMA-positive model. In the case of the combined stimulation with 5 µM 5-aza-dC and 3 mM VPA, the Lu-177–PSMA-617 uptake experiments reveal increased values by a factor of 21 for PC3-PSMA cells compared to the unstimulated cells. These findings are comparable to that obtained for SST2 receptor stimulation in HEKsst_2_ cells [16]. The Lu-177–PSMA uptake enhancement by stimulation of LNCaP cells yields in a factor of 4.7.

Our findings show about three-fold higher Lu-177–PSMA-617 uptake values for unstimulated LNCaP than for unstimulated PC3-PSMA cells (4.8 ± 0.47 kBq vs. 1.4 ± 0.64 kBq), contrary to the results of Liolios et al., who found similar Ga-68–PSMA-11 uptake levels for LNCaP and PC3-PSMA cells [22]. However, these differences might be due to the use of different radioligands and a differing uptake protocol.

The uptake kinetic data obtained for the unstimulated PCa cell lines reveal similar curve shapes for both PC3-PSMA and LNCaP cells. Maximum uptake values are found at 2 h or 4 h for LNCaP or PC3-PSMA, respectively. At 0.5 h and 2 h, LNCaP shows higher uptake values than PC3-PSMA, which is consistent with our results found for unstimulated cell lines in the radioligand uptake studies. By contrast, the kinetic data for Ga-68–PSMA-11 reported by Liolios et al. detects the maximum ligand uptake at around 45 min, while after this time point, cell-bound radioactivity remains the same until 90 min. In terms of the kinetic characteristics, the disagreement with our findings is mainly caused by the chemical different ligands, as well as the different radionuclides (Ga-68–PSMA-11 vs. Lu-177–PSMA-617). In accordance to our findings, the cell-bound radioactivity of LNCaP cells show higher levels than those of PC3-PSMA cells [22]. As it is expected that the enhanced radioligand uptake leads to increased cytotoxic effects, in a next step this hypothesis has to be proven.

Overall, several investigations on antitumor activity on PCa cell lines have been reported [23,24], while studies on epigenetic stimulation of PSMA expression are very limited to date. Sayar et al. recently published investigations on epigenetically mediated PSMA expression in vitro and in vivo. The authors found PSMA decrease in tumor tissues caused by epigenetic changes of the FOLH1 gene, but treatment with a HDACi revoked this epigenetic downregulation and restored PSMA expression in vivo and in vitro [25].

Currently, studies on the upregulation of the PSMA expression are under way to improve the PSMA-targeted therapy. Advanced approaches for target enhancement are not only possible by PSMA-targeting nanocarriers [26] or local application of an I-125-labeled PSMA ligand [27], but also by addressing two cell surface proteins resulting in enhancement of cell-bound radioactivity [22] or increased cytotoxic effects [28].

Recently, investigations in epigenetic targeting of prostate cancer include novel approaches such as combined treatments using epigenetic and hormonal or chemotherapeutic therapies. A combinatorial strategy of epigenetic and hormonal therapies may be a promising perspective in treating advanced prostate cancer [29]. Meller et al. could stimulate the PSMA expression in vitro by a short-term androgen deprivation treatment [30]. He et al. reviewed therapeutic approaches for PCa based on mechanisms targeting e.g., androgen signaling, bone microenvironment, DNA repair pathways, immune checkpoints, cell cycle, and epigenetic modifications [31]. Additionally, the relationship between the methylated histone H3 and development of castration resistant PCa in chemotherapy (docetaxel)-resistant tumor cells has been investigated [32]. Liu et al. found that curcumin at least partly sensitizes prostate cancer to X-ray radiation via epigenetic-mediated mechanisms [33].

In addition to prostate cancer, PSMA could also be upregulated on newly formed tumor vessels of a wide variety of other solid tumors [34], as shown by immunochemistry studies. Hence, in a review by Uijen et al., the PSMA expressions in other solid cancers, especially in the neovasculature, have been described, suggesting a perspective towards a widespread use of PSMA-targeting radioligand therapy in clinical applications [35]. Importantly, the present approach is not limited to radionuclide therapy, but may also apply to immunotherapeutic applications [36], guided surgery, or photodynamic therapy [37].

Overall, our findings reveal an increased Lu-177–PSMA-617 uptake mediated by PSMA expression for both PC3-PSMA and LNCaP cell lines. Enhancing the PSMA expression can be an option to improve the therapeutic efficacy of radionuclide therapy, particularly for patients suffering from mCRPC. Nevertheless, further studies are required to investigate a potentially impaired tumor cell survival caused by upregulation of the PSMA target.

## 4. Materials and Methods

### 4.1. Cell Culture

The human prostate cancer cell line PC3 was obtained from the American Type Culture Collection (ATCC) and served as PSMA-negative control cell. PC3-PSMA cells were obtained from the Helmholtz-Zentrum Dresden-Rossendorf, Institute of Radiopharmaceutical Cancer Research, Dresden, Germany. With the FACS analysis, PSMA expression was detected on LNCaP but not on PC3 cells [36]. To achieve PSMA-expressing PC3 cells, the cells were modified by transducing them with the open reading frame (orf)-encoding prostate stem cell antigen (PSMA). Transduction was performed using a lentiviral packaging system as described previously by Feldmann et al. [38]. LNCaP cells endogenously expressing PSMA were not transduced with the orf-encoding PSMA. All cell lines were cultured in RPMI 1640 medium completed with 10% fetal calf serum (FCS), 1% (*v*/*v*) non-essential amino acids (NEA) (Biochrom, Berlin, Germany). Cells were maintained at 37 °C in a humidified atmosphere of 5% CO_2_.

### 4.2. Immunohistochemistry

The immunohistochemical staining was performed to verify the PSMA expressions on cell membranes in the cell lines PC3, PC3-PSMA, and LNCaP using the Alexa Fluor^®^488 anti-human PSMA (FOLH1) antibody (BIOLEGEND, San Diego, CA, USA).

For immunostaining, cells were grown on chamber slides (Thermofisher scientific, Schwerte, Germany) to 50,000 cells per chamber. Subsequently, cells were fixed in neutral buffered 1% (*v*/*v*) formaldehyde (Merck, Darmstadt, Germany) for 15 min at room temperature. All washing steps were performed with phosphate-buffered saline (PBS). After fixation and three PBS washing steps, the cells were incubated with Alexa Fluor^®^ 488 anti-human PSMA for 1 h at room temperature. Then, 4,6′-diamidino-2-phenylindole (DAPI, Sigma-Aldrich, 0.1 μg/mL, 10 min) was added for staining the cell nuclei. Slides were washed again two times and mounted with fluorescent mounting medium (Dako, Hamburg, Germany). Image analysis was performed using microscope AxioObserverZ.1 (Zeiss AG, Jena, Germany).

### 4.3. Chemical Modulation by 5-aza-dC and VPA

For all stimulation experiments, 5-aza-2′-deoxycytidin (5-aza-dC) and valproic acid sodium salt (VPA) were used (Merck KGaA, Darmstadt, Germany). To start, 5-aza-dC or VPA were dissolved in distilled water at stock solutions of 0.15 mM or 0.5 M, respectively. The chosen concentrations for the experiments were prepared with dilutions in cell culture medium under sterile conditions. To study the effect of 5-aza-dC and VPA on Lu-177–PSMA-617 uptake, the PCa cell lines were plated in T75 flasks and both epidrugs were added alone or in combination on day one. All cell lines were incubated in RPMI 1640 nutrient-deficient medium (lacking medium additives and FCS) until day three at 37 °C. A drug-supplemented deficient medium was changed on day three; thereafter, cells were incubated for another two days in full medium (4.1.). On day five, cells were plated in six-well plates for uptake studies. For the single stimulations, 1.0 or 5.0 µM 5-aza-dC as well as 1.0 mM or 3.0 mM VPA were used. To monitor the combined treatments the following concentrations of 5-aza-dC/VPA were used: 1 µM/1 mM and 5 µM/3 mM.

### 4.4. Radiosynthesis of Lu-177–PSMA-617

The radionuclide Lu-177 (LuCl_3_, non-carrier added, specific activity of 4110 GBq/mg) was provided by the company ITM GmbH (Isotopen Technologies, Muenchen, Germany). The radioactive labelling of PSMA-617 (ABX GmbH, Radeberg, Germany) was performed in a reaction buffer (sodium acetate, gentisic acid) with 10 µg PSMA-617 and 500 MBq Lu-177 at 95 °C for 30 min. The radiochemical product purity was determined with both instant thin-layer chromatography (ITLC) and high-performance liquid chromatography (HPLC). The radiochemical purities were ≥95% for all syntheses.

### 4.5. Intracellular Lu-177–PSMA-617 Radionuclide Uptake and Uptake Kinetic

To examine the PSMA receptor expressions in PC3, PC3-PSMA, and LNCaP cells, the internalized fraction of Lu-177–PSMA was measured after the five day pre-incubation with the described concentrations of 5-aza-dC and VPA, including the samples without the addition of 5-aza-dC or VPA (unstimulated samples). After the stimulation process, 2.5 × 10^5^ cells were distributed as triplicates in six-well MTPs for uptake measurements using a full medium (without 5-aza-dC or VPA). The uptake measurements were performed with 50 kBq Lu-177–PSMA-617 per 1 mL cell culture medium per each well over incubation periods of 1 h or 24 h at 37 °C. For the uptake kinetic experiments, only unstimulated samples were incubated with Lu-177–PSMA-617 at time periods of 0.5, 2, 4, 24, and 48 h under the same conditions as the radionuclide uptake. After each incubation time, the supernatant of the well was removed and the cells were washed twice with PBS at 4 °C. Supernatant and wash solution make up the non-bound fraction of the radioligand. The cell layers were lysed with 1 mL 0.1 M NaOH for 10 min at 37 °C. These fractions correspond to the intracellular fraction of the radioligand. The activity of all fractions was measured using a gamma counter (Cobra II, Autogamma, Packard^®^, Canberra Company, Ruesselsheim, Germany). The cell-bound activity was normalized to the cell count of 2.5 × 10^5^ for all cell lines.

### 4.6. Statistical Analysis

All results of the uptake measurements show the average values and the standard deviation (SD) or the standard error of the mean (SEM) based on two independent trials, in which each experimental condition was performed in triplicate. To examine the statistical significance, an unpaired Student’s *t*-test was used. A difference between two independent samples is significant if the probability of error *p* ≤ 0.05. For the statistical evaluation, GraphPad Prism9 was used.

## 5. Conclusions

Our study demonstrates that the Lu-177–PSMA-617 uptake is increased by a factor of 21 for stimulated PC3-PSMA cells compared to the unstimulated cells. In addition, the PSMA expressions of the genetically modified cell line PC3-PSMA are stimulated in a comparable magnitude to the endogenously PSMA-expressing cell line LNCaP. From that perspective, the present study might contribute to a wide field of advanced therapy approaches in oncology, particularly combining the different treatment steps and complexes such as transfection, stimulation and therapy.

## Figures and Tables

**Figure 1 pharmaceuticals-16-00538-f001:**
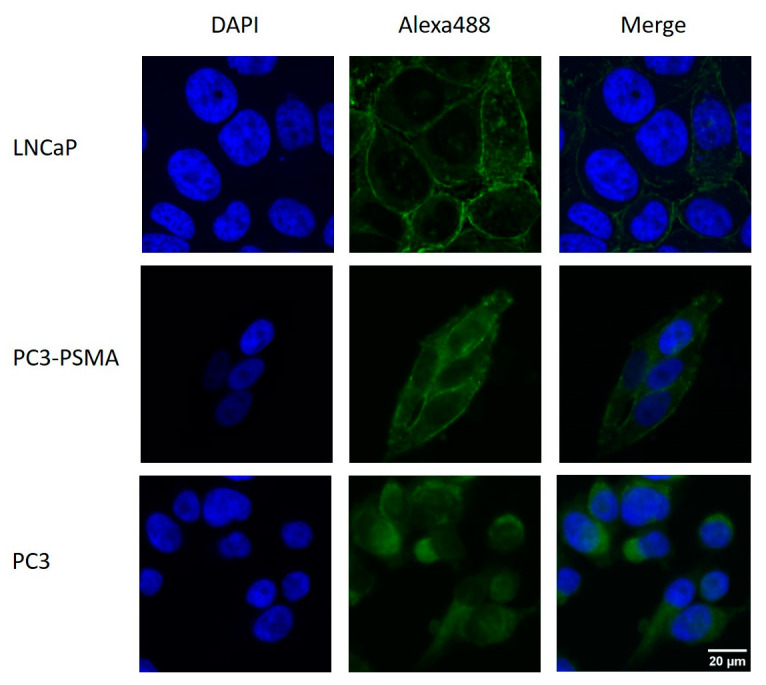
Examples of representative images of PSMA expressions in LNCaP, PC3-PSMA, and PC3 cells. Images are made on AxioObserverZ.1 microscope, 40× magnification.

**Figure 2 pharmaceuticals-16-00538-f002:**
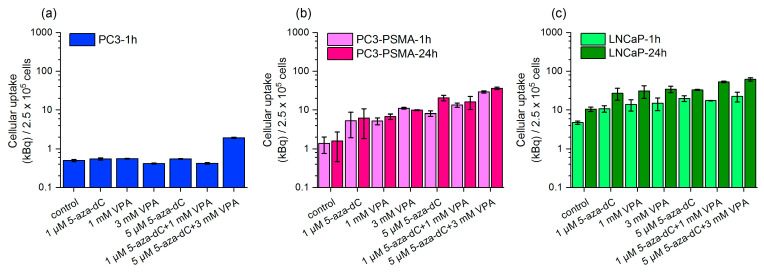
Total cell-bound Lu-177–PSMA-617 (50 kBq/mL radioactive solution) (**a**) for 1 h in PC3 cells, for 1 h and 24 h in (**b**) PC3-PSMA, and (**c**) LNCaP cells after stimulation by 5-aza-dC and VPA. Unstimulated cells (control) were simultaneously incubated with Lu-177–PSMA-617. Data show the average ± SEM.

**Figure 3 pharmaceuticals-16-00538-f003:**
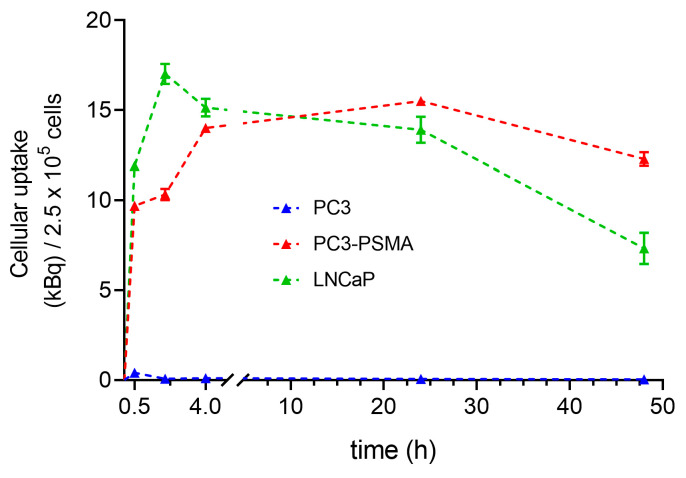
Kinetic data of uptake measurements in PC3, PC3-PSMA, and LNCaP cells at predetermined time points (0.5, 2, 4, 24, and 48 h) after incubation with Lu-177–PSMA-617. Results are displayed as internalized radioactivity (mean ± SD) normalized to 2.5 × 10^5^ cells.

## Data Availability

Data can be requested by contacting the corresponding author.

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
