# Peer review of "Up-Regulation of PSMA Expression In Vitro as Potential Application in Prostate Cancer Therapy"

_pharmaceuticals, 2023, doi:10.3390/ph16040538_

Round 1

Reviewer 1 Report

The paper is well written. It’s aim is clear. The results are practically valuable. Though prostate cancer stimulation with 5-aza-2’-deoxycitidine (5-aza-dC) and valproic acid (VPA) for radionuclide therapy with PSMA radiopharmaceuticals is very far from clinical application so far, the study is interesting as a first step to clinic as well as possible useful instrument for preclinical and translational studies.

Some specific issues should be addressed:

1.                  Radioligand therapy (RLT) should be replaced by Radionuclide Therapy as a standard name of this method

2.                  Columns in Fig 2(a) are not visible. It is understandable that the signal was very low for PC3, but maybe logarithmic scale should be used make them visible along with nearby diagrams.

3.                  Line 196 – Authors should avoid speculations on reduction of injected radiopharmaceuticals dose and thus sparing kidneys. Cancer is to crucial and strong enemy to fight. Currently strategy AHASA  (As High as Safely Administrable) is used in Radionuclide Therapy.

4.                  What is the role and purpose of intracellular uptake of Lu-177-PSMA- 617 in UNSTIMULATED cells study?

5.                  Conclusion section is absent.

Reviewer 2 Report

The short commentary reveals a study that investigated the stimulation effect of  (5-aza-dC) and valproic acid (VPA) on prostate cancer cell lines. The results are interesting and potentially lead to generating hypotheses that impact prostate cancer treatment.

Overall, the article is well-written, and easy to follow the content. I found in Page 6, line 247 the following information:

"For the single stimulations 1.0 or 5.0 μM 5-aza-dC as well 247 as 1.0 mM or 3.0 mM VPA were used. To monitor the combined treatments the following 248 concentrations of 5-aza-dC / VPA were used: 1 μM / 1 mM and 5 μM / 3 mM, respectively."

my question is: what are the reasons for the selection of different concentrations here?

Another optional suggestion:

It would be more informative if the authors could provide additional information on how they determined the solution concentration for optimal cellular uptakes.
